# Intervention Design for Causal Representation Learning

**Phillip Lippe**[1]    **Sara Magliacane**[2,3]    **Sindy Löwe**[4]    **Yuki M. Asano**[1]    **Taco Cohen**[5]    **Efstratios Gavves**[1]

[1]QUVA Lab, University of Amsterdam
[2]MIT-IBM Watson AI Lab
[3]INDE Lab, University of Amsterdam
[4]UvA-Bosch Delta Lab, University of Amsterdam
[5] Qualcomm AI Research[*], Amsterdam, Netherlands

## Abstract

In this paper, we take a first step towards bringing two fields of causality closer together: intervention design and causal representation learning. Intervention design is a well studied task in classic causal discovery, which aims at finding the minimal sets of experiments under which the causal graph can be identified. Causal representation learning aims at recovering causal variables from high-dimensional entangled observations. In recent work in causal representation, interventions are exploited to improve identifiability, similarly to classic causal discovery. Hence, the same task becomes relevant in this setting as well: how many experiments are minimally needed to identify the latent causal variables? Based on the recent causal representation learning method CITRIS, we show that for $K$ causal variables, $\lfloor \log_2(K) \rfloor + 2$ experiments are sufficient to identify causal variables from temporal, intervened sequences, which is only one more experiment than needed for classic causal discovery in the worst case. Further, we show that this bound holds empirically in experiments on a 3D rendered video dataset.

## 1 INTRODUCTION

Recently, there has been a growing interest in the field of causal representation learning (Brehmer et al., 2022; von Kügelgen et al., 2021; Lippe et al., 2022a,b; Locatello et al., 2020; Schölkopf et al., 2021), which aims at discovering latent, causal factors and their causal relations from high-dimensional observations such as images or videos. A crucial aspect towards reaching this goal is commonly considered to be the access to interventional data. An intervention influences the causal mechanism of one or more vari-

ables while leaving other mechanisms and the observation function, *i.e.* the way how we perceive these variables, unchanged. Comparing data under different interventions allows one to find and disentangle the sources of variations, *i.e.* the causal variables, from the high-dimensional observations. In the context of causal representation learning, several works have considered different interventional settings, such as interventions with unknown targets (Brehmer et al., 2022; Locatello et al., 2020; Yao et al., 2022) or observed targets (von Kügelgen et al., 2021; Lachapelle et al., 2022; Lippe et al., 2022b), in order to guarantee identifiability of the causal variables. However, interventional data can often be expensive, since it requires a specific experiment in which there is a perturbation of the causal system, e.g. a randomized controlled trial. Hence, a naturally arising question is what is the minimal number of different intervention experiments that suffices for identifying the causal variables.

In this paper, we answer this question by drawing connections from causal representation learning to the area of intervention design (Eberhardt, 2007; Greenewald et al., 2019; He and Geng, 2008; Hyttinen et al., 2012a, 2013; Kocaoglu et al., 2017a,b; Shanmugam et al., 2015; Squires et al., 2020), which aims to find the minimal set of experiments that identifies the causal graph for known causal variables. As a specific setting, we focus on the recent causal representation learning method CITRIS (Lippe et al., 2022b) which leverages data from temporal intervened sequences with known intervention targets. Using similar techniques as for intervention design, we show that for CITRIS $\lfloor \log_2(K) \rfloor + 2$ experiments are sufficient to identify the $K$ causal variables, which is exactly one experiment more than needed for causal discovery in the worst case scenario. This opens up several opportunities for future work to leverage intervention design methods from causal discovery also in causal representation learning.

Furthermore, to show that this bound holds also empirically, we conduct experiments on the Temporal Causal3DIdent dataset (von Kügelgen et al., 2021; Lippe et al., 2022b). This dataset contains videos of 3D object renderings with

---

[*] Qualcomm AI Research is an initiative of Qualcomm Technologies, Inc.

*Accepted for the Causal Representation Learning workshop at the 38*[th] *Conference on Uncertainty in Artificial Intelligence* (UAI CRL 2022).

6 causal variables, including the object positions, rotations, colors, and lightning. As expected based on our theoretical results, with a minimal set of $\lfloor \log_2(6) \rfloor + 2 = 4$ experiments, CITRIS is able to disentangle the variables well, obtaining an only slightly higher disentanglement error compared to the same model with access to an unlimited number of intervention experiments.

## 2 RELATED WORK

### 2.1 INTERVENTION DESIGN

Intervention design, also known as experiment selection, is a well-studied area in causal discovery (Addanki et al., 2020; Eberhardt, 2007; Eberhardt et al., 2005; Eberhardt and Scheines, 2007; Hauser and Bühlmann, 2014; He and Geng, 2008; Hyttinen et al., 2012a, 2013; Kocaoglu et al., 2017a,b; Shanmugam et al., 2015; Squires et al., 2020). Consider an unknown causal graph $G = (V, E)$, where each node $i \in V$ is associated with a causal variable $C_i$, and each edge $(i, j) \in E$ represents a causal relation from $C_i$ to $C_j$: $C_i \to C_j$. Given a set of $K$ causal variables $C_1, ..., C_K$, the goal is to determine the set of experiments $\{E_1, ..., E_N\}$ which identifies the underlying causal structure $G$ most efficiently. In our notation, an experiment $E_i$ is defined as a set of causal variables on which interventions are jointly performed. For example, the observational regime is denoted by $E_i = \emptyset$, *i.e.* all variables are passively observed, and single-target interventions by $E_1 = \{C_1\}, E_2 = \{C_2\}$, and so on.

One can show that under causal sufficiency (*i.e.*, no latent confounders or selection bias), acyclicity (*i.e.*, no cycles in $G$), and the causal Markov and faithfulness assumptions (*i.e.*, there are no additional independences w.r.t. the ones encoded in the graph $G$), a set of experiments uniquely identifies the causal graph if for every pair of variables $(C_i, C_j)$, (1) there exists an experiment where $C_i$ has been intervened on, but not $C_j$, or vice versa, and (2) there exists an experiment under which both $C_i$ and $C_j$ are passively observed, *i.e.* not intervened. The first condition is also referred to as the *unordered pair condition*, and the latter as the *covariance condition* (Eberhardt, 2007; Hyttinen et al., 2013). Using these conditions, Eberhardt (2007) showed that $\lfloor \log_2(K) \rfloor + 1$ experiments are sufficient to guarantee the identifiability of a causal graph with $K$ variables. This bound is for the worst case scenario, and for certain graphs, fewer experiments might be sufficient. For instance, if all variables are independent, the observational regime alone identifies the whole graph.

While in this work we provide a first application of intervention design for causal representation learning, in the intervention design literature, several other settings can be considered, including (1) choosing the experiments adaptively after seeing the results of the previous ones (Greenewald et al., 2019; Hauser and Bühlmann, 2014; He and Geng,

2008; Squires et al., 2020), (2) the number of variables that can be subject to an intervention simultaneously being limited (Ghassami et al., 2018; Hyttinen et al., 2013; Kocaoglu et al., 2017a; Shanmugam et al., 2015), (3) that latent confounders may exist (Addanki et al., 2020; Hyttinen et al., 2012b; Kocaoglu et al., 2017b), or (4) that additional background knowledge about the underlying causal structure is available (Eberhardt, 2008; Greenewald et al., 2019; Hauser and Bühlmann, 2012; Sen et al., 2017).

### 2.2 CAUSAL REPRESENTATION LEARNING

Causal representation learning aims at learning representations of causal factors in an underlying system from high-dimensional observations like images (Brehmer et al., 2022; von Kügelgen et al., 2021; Lachapelle et al., 2022; Lippe et al., 2022a,b; Locatello et al., 2020; Schölkopf et al., 2021). One of the first lines of work is Independent Component Analysis (ICA) (Comon, 1994; Hyvärinen et al., 2001) trying to recover independent latent variables entangled by some invertible transformation. ICA was extended to nonlinear transformations by exploiting auxiliary variables under which the latents become conditionally mutually independent (Hyvärinen and Morioka, 2016; Hyvärinen et al., 2019), and combined with deep learning architectures like VAEs (Khemakhem et al., 2020a,b; Sorrenson et al., 2020; Zimmermann et al., 2021). Further, recent works draw a connection between causality and ICA (Gresele et al., 2021; Monti et al., 2019). In particular, Lachapelle et al. (2022); Yao et al. (2022) discuss identifiability from temporal sequences. While both can model interventions in their framework, they do not explicitly exploit the knowledge of the intervention targets and require additional assumptions in terms of sufficient variation.

Focusing on causal structures in the data, von Kügelgen et al. (2021) demonstrate that contrastive learning methods can block-identify causal variables by considering augmentations as interventions on the style of the image, while keeping the content unchanged. Locatello et al. (2020) identify independent latent causal factors from pairs of observations that only differ in a subset of causal factors. Brehmer et al. (2022) have recently extended this setup to variables with instantaneous causal effects, but require pairs of observations that share the noise term for all variables except one intervened variable, *i.e.* counterfactual samples. Finally, CITRIS (Lippe et al., 2022b) uses temporal sequences with interventions to identify the minimal causal variables, *i.e.* the part of a potentially multidimensional causal variable that is influenced by the provided interventions. Thereby, CITRIS considers the causal variables to have temporal dependencies, but being independent within a time step conditioned on the previous time step. In this paper, we focus on the relation of CITRIS (Lippe et al., 2022b) to intervention design, since its setting is the closest to common intervention de-

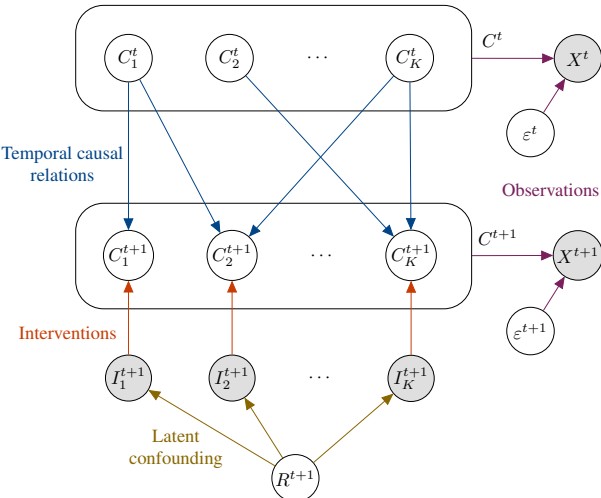

Figure 1: An example causal graph in CITRIS. A latent causal factor $C_i^{t+1}$ has as parents a subset of the causal factors at the previous time step $C_1^t, \ldots, C_K^t$, and its intervention target $I_i^{t+1}$. All causal variables $C^{t+1}$ and the noise $\varepsilon^{t+1}$ cause the observation $X^{t+1}$. $R^{t+1}$ is a latent confounder between the intervention targets.

sign methods by explicitly considering intervention targets for $K$ variables. Still, we note that similar derivations may be possible for other causal representation learning methods, like Lachapelle et al. (2022), under additional assumptions.

# 3 INTERVENTION DESIGN FOR CAUSAL REPRESENTATION LEARNING

We first review the setting and assumptions on which CITRIS (Lippe et al., 2022b) aims to identify the causal variables from high-dimensional observations. For simplicity, we limit ourselves to the aspects relevant for designing the intervention experiments, and refer to Lippe et al. (2022b) for the full details. We then discuss the identifiability conditions of CITRIS with respect to the interventions, and bring them into context of common intervention design settings. Finally, we derive the minimal number of experiments that are sufficient to fulfill these conditions, and show a simple heuristic with which we can create such minimal sets.

## 3.1 TEMPORAL INTERVENED SEQUENCES

CITRIS considers a latent, temporal dynamical system with $K$ causal variables, $(C_1^t, C_2^t, ..., C_K^t)_{t=1}^T$, as visualized in Figure 1. The causal graph may contain temporal relations, but no instantaneous effects. This means that we can model each causal factor by $C_i^t = f_i(\text{pa}_G(C_i^t), \epsilon_i)$ for $t = 1, \ldots, T$ and $i = 1, \ldots, K$, where $\text{pa}(C_i^t) \subseteq \{C_1^{t-1}, \ldots, C_K^{t-1}\}$. The graph structure is thereby time in-

variant, *i.e.*, $\text{pa}_G(C_i^t) = \text{pa}_G(C_i^1)$ for any $t = 1, \ldots, T$, and all $\epsilon_i$ are mutually independent noises.

Instead of observing the causal variables directly, we measure high-dimensional observations $X^t$ at each time step $t$. These observations represent an entangled view of all causal factors, $C^t$, and possible noise variables $\varepsilon^t$.

To represent interventions, we augment the graph with binary intervention target variables $I^t \in \{0, 1\}^K$, where $I_i^t = 1$ if $C_i^t$ has been intervened upon, and $I_i^t = 0$ otherwise. In the causal graph, each intervention variable $I_i^t$ is a parent of its respective causal variable $C_i^t$. Thereby, CITRIS supports soft interventions (Eberhardt, 2007), which change the conditional distribution, *i.e.*, $p(C_i^t|\text{pa}(C_i^t), I_i^t = 1) \neq p(C_i^t|\text{pa}(C_i^t), I_i^t = 0)$, and includes *perfect* interventions $\text{do}(C_i = c_i)$ (Pearl, 2009) as a special case. In this paper, we consider the general case where interventions could be either soft, perfect, or mixed across variables. Further, we assume that the form of intervention, *i.e.* the way an intervention changes an existing mechanism, can be arbitrary and externally determined. To model dependencies between interventions, CITRIS considers a potential latent confounder $R^t$ between the intervention targets. In alignment with the intervention design setting, $R^t$ could represent the index of the experiment $E_i$ which has been performed at time step $t$.

Moreover, the causal variables are considered to be potentially multidimensional, *e.g.* representing a 3d position as $C_i \in \mathbb{R}^3$. As shown by Lippe et al. (2022b), multidimensional causal variables are not always identifiable from interventions, since interventions may only affect a subset of the variables dimensions. Instead, we split a causal variable into two parts, $s_i(C_i^t) = (s_i^{\text{var}}(C_i^t), s_i^{\text{inv}}(C_i^t))$ where $s_i$ is invertible. In this split, $s_i^{\text{var}}(C_i^t)$ represents the *intervention-dependent* part of $C_i$ and $s_i^{\text{inv}}(C_i^t)$ the intervention-independent part. We can then write the distribution of a causal variable as:

$$p\left(s_i(C_i^{t+1})|C^t, I_i^{t+1}\right) = p\left(s_i^{\text{var}}(C_i^{t+1})|C^t, I_i^{t+1}\right) \cdot p\left(s_i^{\text{inv}}(C_i^{t+1})|C^t\right) \quad (1)$$

Intuitively, for a multidimensional variable where an intervention may only affect a subset of dimensions, $s_i^{\text{var}}(C_i^{t+1})$ represents the dimensions that depend on the intervention, and $s_i^{\text{inv}}(C_i^{t+1})$ the dimensions whose mechanisms are invariant to interventions. Under this setting, Lippe et al. (2022b) define a *minimal causal variable* as follows:

**Definition 3.1.** *The* minimal causal variable *of a causal variable $C_i$ with respect to its intervention variable $I_i$ is $s_i^{\text{var}}(C_i^t)$ of the split $s_i(C_i^t) = (s_i^{\text{var}}(C_i^t), s_i^{\text{inv}}(C_i^t))$, which maximizes the information content $H(s_i^{\text{inv}}(C_i^t)|pa(C_i^t))$.*

In other words, the minimal causal variable is the smallest part of a causal variable that strictly depends on the provided intervention. In practice, the minimal causal variable is commonly the same as the original causal variable if an

intervention affects all its dimensions, *e.g.* an intervention on the position of an object changes the mechanism of the $x, y,$ and $z$ coordinate simultaneously.

## 3.2 CONDITIONS FOR IDENTIFYING MINIMAL CAUSAL VARIABLES

Under the previously described setting, Lippe et al. (2022b) show that the minimal causal variable of a causal factor $C_i$ is identifiable if for any other variable $C_j, i \neq j$, the following condition holds:

$$C_i^{t+1} \not\perp\!\!\!\perp I_i^{t+1} | C^t, I_j^{t+1} \qquad (2)$$

Intuitively, we cannot disentangle the causal variables based on interventions if the intervention target variable $I_i^{t+1}$ is constant or can be replaced by any other target variable $I_j^{t+1}$. To make it more explicit, Equation (2) can be rewritten as the following four conditions:

1. there exists a time step $\tau$ such that $C_i$ is intervened, *i.e.*, $I_i^\tau = 1$;
2. there exists another time step $\tau'$ such that $C_i$ is not intervened, *i.e.*, $I_i^{\tau'} = 0$;
3. for all other causal factors $C_j$, there exists a time step $\tilde{\tau}$ when $I_i^{\tilde{\tau}} \neq I_j^{\tilde{\tau}}$;
4. for all other causal factors $C_j$, there exists a time step $\hat{\tau}$ when $I_i^{\hat{\tau}} = I_j^{\hat{\tau}}$.

The first two conditions ensure that $I_i$ is not constant, which is strictly necessary for Equation (2) to hold. The last two conditions guarantee that if the intervention target $I_i$ is not constant, then there cannot exist a deterministic function $f$ for which $I_i^\tau = f(I_j^\tau)$ for all time steps $\tau$. This is sufficient for fulfilling Equation (2), since $I_i^{t+1}$ cannot be determined from $I_j^{t+1}$ and $C^t$ and we assume that the minimal causal variable is not the empty set, *i.e.* $C_i^{t+1} \not\perp\!\!\!\perp I_i^{t+1}$ without conditions. At the same time, these conditions are necessary, because if condition 3 or 4 does not hold, we have $I_i^{\hat{\tau}} = I_j^{\hat{\tau}}$ or $I_i^{\hat{\tau}} \neq I_j^{\hat{\tau}}$ for *all* time steps $\tau$. This means that there exists a deterministic function with $I_i^\tau = f(I_j^\tau)$, and hence $C_i^{t+1} \perp\!\!\!\perp I_i^{t+1} | I_j^{t+1}$, violating Equation (2). Hence, conditions 1-4 are both sufficient and necessary with respect to Equation (2).

Instead of considering different time steps, we can also reformulate the conditions in terms of different experiments. Any distribution over intervention targets, $p(I^{t+1})$, can be represented by a distribution over all possible combinations of intervention targets, *i.e.* the experiments $E_1, ..., E_N$. In this case, the regime variable $R^{t+1}$ can be seen as an experiment indicator, where $R^{t+1} = l$ denotes that the intervention targets at time step $t + 1$ follow experiment $E_l$. Thus, the distribution $R^{t+1}$ follows the distribution over the experiments. For a limited sample size, this distribution may correspond to the number of samples we have per experiment. On the other hand, the conditional distribution of the intervention targets can then be written as a deterministic

function of the regime variable:

$$p(I_i^{t+1} = 1 | R^{t+1} = l) = \mathbb{1}\left[C_i \in E_l\right] \qquad (3)$$

In other words, $I_i^{t+1} = 1$ if $C_i$ has been intervened on in the experiment $E_l$ (with $R^{t+1} = l$), and otherwise 0. Now, considering a set of the experiments $\{E_1, ..., E_N\}$, we can rewrite the conditions above as follows:

1. there exists an experiment $E_l$ such that $C_i \in E_l$;
2. there exists an experiment $E_o$ such that $C_i \notin E_o$;
3. for all other causal factors $C_j, i \neq j$, there exists an experiment $E_{p_j}$ such that $C_i \in E_{p_j}$ and $C_j \notin E_{p_j}$, or $C_i \notin E_{p_j}$ and $C_j \in E_{p_j}$;
4. for all other causal factors $C_j, i \neq j$, there exists an experiment $E_{q_j}$ such that $C_i, C_j \in E_{q_j}$, or $C_i, C_j \notin E_{q_j}$;

Note that the experiments fulfilling these conditions do not need to be mutually exclusive, *i.e.* an experiment can fulfill both condition 1 and 2. With this, our goal becomes finding the minimal number of experiments $N'$ for a given number of causal variables $K$ that can fulfill these four conditions.

## 3.3 DERIVING THE MINIMAL NUMBER OF EXPERIMENTS

Based on the four conditions, we derive the minimal number of experiments in two steps. Firstly, following condition 3, we need to ensure that every variable $C_i$ has a unique pattern of being intervened or passively observed in different experiments. In other words, the intervention target variables $I_i, I_j$ of two causal variables $C_i, C_j$ cannot be identical for all experiments. For simpler exposition, we express the intervention pattern of a causal variable $C_i$ across $N$ experiments as a binary code $b^i \in \{0, 1\}^N$, where $b_l^i = \mathbb{1}\left[C_i \in E_l\right]$. As an example, consider two variables $C_1, C_2$, for which we have the two experiments $E_1 = \{C_1\}, E_2 = \{C_1, C_2\}$. We represent this pattern with the binary code $b^1 = 11$ for $C_1$, since $C_1$ is intervened in both experiments, and $b^2 = 01$ for $C_2$, since $C_2$ is only passively observed in $E_1$. Essentially, the binary code $b^i$ is a concatenation of the binary intervention targets of a variable $C_i$ across experiments $E_1, ..., E_N$.

With this representation, the condition that the intervention target variables of two variables, $I_i, I_j$, cannot be identical for all experiments, translates to their binary code $b^i, b^j$ being unique, *i.e.* different in at least one digit. To ensure that the binary codes for all variables $b^1, ..., b^K$ are unique, we would need at least $\lceil \log_2(K) \rceil$ experiments, since a binary code of length $L = \lceil \log_2(K) \rceil$ can represent $2^L \geq K$ different values. However, we need to exclude two codes: (1) the code of all zeros $\{0\}^L$, which violates condition 1, and (2) the code of all ones $\{1\}^L$, *i.e.*, interventions at all time, violating condition 2. Removing these two from the available binary codes, we obtain a minimum code length of $\lceil \log_2(K + 2) \rceil$ in order to have $K$ unique, valid codes.

Consider, for example, two variables $C_1, C_2$, for which we have four possible binary codes: $\{00, 01, 10, 11\}$. Removing the code for a variable being always passively observed, $00$, and the code for a variable always being intervened, $11$, we are left with the $\lceil \log_2(2+2) \rceil = 2$ codes $\{01, 10\}$.

In a second step, we need to extend this bound to fulfill condition 4, *i.e.* for all pairs $C_i, C_j$, there exists an experiment for which $I_i = I_j$ holds. In other words, we need to prevent that $b^i = \neg b^j$ for any $i, j$, with $\neg b$ representing the one's complement of a binary code $b$. In the example above, this implies that we cannot use the codes $b^1 = 01$ and $b^2 = 10$ for $C_1, C_2$, since $b^1 = \neg b^2$. More generally, any subset of more than $2^{L-1}$ unique codes of length $L$ must contain a pair $b^i, b^j$ for which $b^i = \neg b^j$. Hence, we effectively need to double the previous number of codes to ensure that there exist a subset of $K$ unique codes that are not complementary to each other. Note, however, that this does not affect the two constant codes, all zeros and all ones, since they are invalid in any code space and are the inverse of each other. Hence, the minimal code length to find $K$ binary codes that are (1) unique, (2) non-constant, and (3) not complementarities of each others, is $\lceil \log_2(2K+2) \rceil = \lceil \log_2(K+1) \rceil + 1 = \lfloor \log_2(K) \rfloor + 2$.[1] This bound simultaneously corresponds to the minimal number of experiments needed to fulfill the conditions 1-4, which we summarize in the following proposition:

**Proposition 3.2.** *The minimal number of experiments to fulfill the identifiability condition of Equation* (2) *is* $\lfloor \log_2(K) \rfloor + 2$, *with $K$ being the number of causal variables.*

One way of preventing codes from being complementary to each other is to add the purely observational regime, *i.e.* $E_0 = \emptyset$, to an experimental set that already fulfills conditions 1-3. This ensures that there exist an experiment $E_l$ for which $C_i, C_j \notin E_l$ and thus $b_l^i = b_l^j$ for all pairs.

A simple algorithm for creating such sets of experiments is shown in Algorithm 1. Taking again the example of two variables $C_1, C_2$, we first create all binary codes of length $\lfloor \log_2(2) \rfloor + 1 = 2$: $\mathcal{B} = \{00, 01, 10, 11\}$ (*i.e.* one experiment less than the bound, since we add the observational experiment later). We then remove the code for passively observing a specific causal variable in all experiments: $\mathcal{B} \setminus \{00\} = \{01, 10, 11\}$. Next, we add the experiment in which *all* causal variables are jointly, passively observed: $\mathcal{B} = \{001, 010, 011\}$. From these three codes, we pick two codes for the two causal variables $C_1, C_2$. Note that any combination of two codes from $\mathcal{B}$ is valid, and we could pick them based on some heuristic, for instance minimizing the number of interventions: $b^1 = 001$, $b^2 = 010$. We then create the $\lfloor \log_2(2) \rfloor + 2 = 3$ experiments based on these codes: $E_1 = \emptyset, E_2 = \{C_2\}, E_3 = \{C_1\}$.

---

[1] These equalities are possible, since we have $K \in \mathbb{N}$, *i.e.* the number of causal variables is a positive integer greater than zero.

---

**Algorithm 1** Pseudocode for finding a minimal set of experiments that enable the identification of the minimal causal variables with an observational regime.

---
**Require:** Number of variables $K$
1: Create all possible binary codes of length $L = \lfloor \log_2(K) \rfloor + 1$ as set $\mathcal{B} = \{0, 1\}^L$
2: Remove the code of observing a variable passively in all experiments, $\{0\}^L$, from $\mathcal{B}$
3: Extend all codes in $\mathcal{B}$ by appending $\{0\}$, *i.e.* an experiment where all variables are passively observed
4: From the remaining codes in $\mathcal{B}$, (arbitrarily) pick $K$ unique codes $b^1, ..., b^K$, one for each causal variable $C_i$
5: Create experiments by using the codes as binary intervention targets: $E_l = \{C_i | i \in [\![1..K]\!], b_l^i = 1\}$

---

In conclusion, we have shown that we can guarantee to find the minimal causal variables of a set of causal variables $C_1, ..., C_K$ with as little as $\lfloor \log_2(K) \rfloor + 2$ experiments, of which one can always be the observational regime. Furthermore, the results generalize to iCITRIS (Lippe et al., 2022a), a recent extension of CITRIS to instantaneous effects, when considering perfect interventions, since both rely on the same intervention condition of Equation (2). In comparison to the bound derived by Eberhardt (2007) for causal discovery ($\lfloor \log_2(K) \rfloor + 1$) in the worst case scenario, we require just exactly one additional experiment to identify the minimal causal variables. Despite the different setups and goals in causal representation learning and causal discovery, the similarity of the two bounds suggests that we can potentially use similar extensions of the causal discovery domain, with minimal adjustments, for causal representation learning, since we may only have to add one more experiment. Such extensions include, for example, limiting the number of simultaneous interventions (Hyttinen et al., 2013) or selecting the cheapest set of experiments according to some cost function (Ghassami et al., 2018; Kocaoglu et al., 2017a; Lindgren et al., 2018).

## 4 EXPERIMENTS

To verify that CITRIS can operate in a limited experimental setting, we repeat the experiments of Lippe et al. (2022b) on the Temporal Causal3DIdent dataset, but with a smaller set of interventions. The Temporal Causal3DIdent dataset consists of 3D renderings ($64 \times 64$ pixels) of an object shape under varying positions, rotations, and lights. For simplicity, we fix the shape to a teapot, which leaves six causal variables that causally interact over time. Using our bound derived in Section 3.3, we obtain that $\lfloor \log_2(6) \rfloor + 2 = 4$ intervention experiments are sufficient to identify the variables of the Temporal Causal3DIdent Teapot dataset. Hence, we sample four intervention experiments following Algorithm 1, and show one example of the experiment set in Table 2. The

Table 1: Results on the Temporal Causal3DIdent dataset with different experimental settings over three seeds. *Full experiments* denotes the setting with full support over all possible intervention experiments (*i.e.*, $I_i^{t+1} \sim \text{Bernoulli}(0.1)$, results taken from Lippe et al. (2022b)), and *minimal experiments* follow minimal sets of experiments (ours). $R^2$ diag and Spearman diag measure the correlation between a causal variable and the latent variables assigned to it by CITRIS. $R^2$ sep and Spearman sep denote the maximum correlation to any other causal variable. The triplet distance measures the disentanglement by testing the generation of new combinations of causal factors (see Lippe et al. (2022b) for detailed descriptions).

| Experimental setting | Triplets ↓ | $R^2$ diag ↑ | $R^2$ sep ↓ | Spearman diag ↑ | Spearman sep ↓ |
|---|---|---|---|---|---|
| iVAE - Full experiments | 0.15 (±0.01) | 0.78 (±0.04) | 0.21 (±0.10) | 0.77 (±0.05) | 0.17 (±0.04) |
| CITRIS - Full experiments | 0.04 (±0.00) | 0.98 (±0.00) | 0.01 (±0.00) | 0.97 (±0.00) | 0.05 (±0.01) |
| CITRIS - Minimal experiments | 0.12 (±0.02) | 0.94 (±0.05) | 0.08 (±0.05) | 0.92 (±0.08) | 0.10 (±0.05) |

Table 2: An example of the selected experiments under the minimal number of experiments setting, generated according to Algorithm 1. $E_1$ is the observational regime, and $E_2, E_3, E_4$ create unique intervention patterns for each causal variable.

| | $E_1$ | $E_2$ | $E_3$ | $E_4$ |
|---|---|---|---|---|
| pos_o | - | ✓ | - | ✓ |
| rot_o | - | - | ✓ | - |
| rot_s | - | - | ✓ | ✓ |
| hue_o | - | ✓ | ✓ | - |
| hue_b | - | - | - | ✓ |
| hue_s | - | ✓ | - | - |

first experiment, $E_1$, is the observational regime where all variables are passively observed, and the other experiments $E_2, E_3, E_4$ cover the needed interventions. The data is generated following the same process as in Lippe et al. (2022b). For the regime variable, we sample the purely observational experiment $E_1$ 50% of the time, and uniformly between $E_2, E_3, E_4$ otherwise. Hence, we obtain samples from all experiments, with a bias towards observational data, since this is usually cheaper to obtain.

In Table 1, we show the results of CITRIS on the Temporal Causal3DIdent dataset under different experimental settings. The setting "*full experiments*" is the original setup of Lippe et al. (2022b), where the intervention targets are independently sampled from a Bernoulli distribution, *i.e.* $I_i^{t+1} \sim \text{Bernoulli}(0.1)$. Hence, we effectively obtain samples from all possible experiments. In contrast, the setting "*minimal experiments*" only uses four experiments, as described before. We repeat all experiments with three different seeds and three different minimal sets of experiments (see Appendix A.1 for the specific sets). The results on the minimal experiment set show that CITRIS is still able to disentangle the causal variables decently, with small degradation in performance compared to the full experiments. In general, we find that the model is more likely to entangle variables with a very similar intervention pattern due to possible local minima. For example, for the set of experiments

in Table 2, one model seed entangled the background hue (hue_b) and the object position (pos_o), which, in terms of interventions, only differ in experiment $E_2$. Still, CITRIS under a minimal set of experiments considerably outperforms the best baseline model, an iVAE (Khemakhem et al., 2020a), on a full set of experiments. In conclusion, CITRIS can identify the causal variables even under a minimal set of experiments well, but it is more challenging to optimize due to strong dependencies between intervention targets.

## 5 CONCLUSION

In this paper, we show that $\lfloor \log_2(K) \rfloor + 2$ intervention experiments are sufficient to identify $K$ causal variables from high-dimensional observations like images for the causal representation learning method CITRIS. This bound has a strong resemblance to the bound in intervention design, which guarantees the discovery of a causal graph for known variables in just one experiment less than in the bound we present. Further, we empirically verify this bound by showing that CITRIS with a minimal set of four experiments disentangles the six causal variables of the Temporal Causal3DIdent dataset almost as well as with an unlimited number of experiments. This suggests that adapting further methods from the field of intervention design to causal representation learning holds promise for future work, and the presented work can provide a first step towards this goal.

**Author Contributions**

P. Lippe and S. Magliacane conceived the idea. P. Lippe developed the theoretical results, created the datasets, ran the experiments, and wrote the paper. S. Löwe and S. Magliacane provided comments and feedback throughout the writing process. S. Magliacane, S. Löwe, Y. M. Asano, T. Cohen, E. Gavves advised during the project.

**Acknowledgements**

We thank SURFsara for the support in using the Lisa Compute Cluster. This work is financially supported by Qual-

comm Technologies Inc., the University of Amsterdam and the allowance Top consortia for Knowledge and Innovation (TKIs) from the Netherlands Ministry of Economic Affairs and Climate Policy.

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

# A EXPERIMENTAL DETAILS

This section provides additional details on the experimental setting. We first show the specific minimal sets of experiments used in the Temporal Causal3DIdent dataset (Appendix A.1). Next, we provide examples of correlation maps for these experiments.

## A.1 MINIMAL SETS OF EXPERIMENTS

The chosen sets of experiments are shown in Figure 2. In each set, the experiment $E_1$ is the observational regime, and $E_2, E_3, E_4$ cover all conditions for the interventions. The assignment of binary codes to causal variables, as performed in Algorithm 1, is done mostly randomly. To cover all critically different settings, we ensure that for each causal vari-

|  | $E_1$ | $E_2$ | $E_3$ | $E_4$ |
|---|---|---|---|---|
| pos_o | - | ✓ | - | ✓ |
| rot_o | - | - | ✓ | - |
| rot_s | - | - | ✓ | ✓ |
| hue_o | - | ✓ | ✓ | - |
| hue_b | - | - | - | ✓ |
| hue_s | - | ✓ | - | - |

(a) Experimental setting 1

|  | $E_1$ | $E_2$ | $E_3$ | $E_4$ |
|---|---|---|---|---|
| pos_o | - | ✓ | - | - |
| rot_o | - | - | ✓ | - |
| rot_s | - | ✓ | ✓ | - |
| hue_o | - | - | - | ✓ |
| hue_b | - | ✓ | - | ✓ |
| hue_s | - | - | ✓ | ✓ |

(b) Experimental setting 2

|  | $E_1$ | $E_2$ | $E_3$ | $E_4$ |
|---|---|---|---|---|
| pos_o | - | ✓ | - | ✓ |
| rot_o | - | - | ✓ | ✓ |
| rot_s | - | - | - | ✓ |
| hue_o | - | - | ✓ | - |
| hue_b | - | ✓ | - | - |
| hue_s | - | ✓ | ✓ | - |

(c) Experimental setting 3

Figure 2: Minimal sets of experiments used in the Temporal Causal3DIdent experiments. "✓" indicates that a causal variable is intervened in the corresponding experiment. For example, in the first experimental setting (Table 2a), pos_o is intervened in $E_2$. "-" denotes a passive observation of a variable in this experiment. The three experimental settings were randomly chosen, whereby $E_1$ is always the observational regime, and each causal variable is at least in one setting only intervened in one experiment, and once intervened in two experiments.

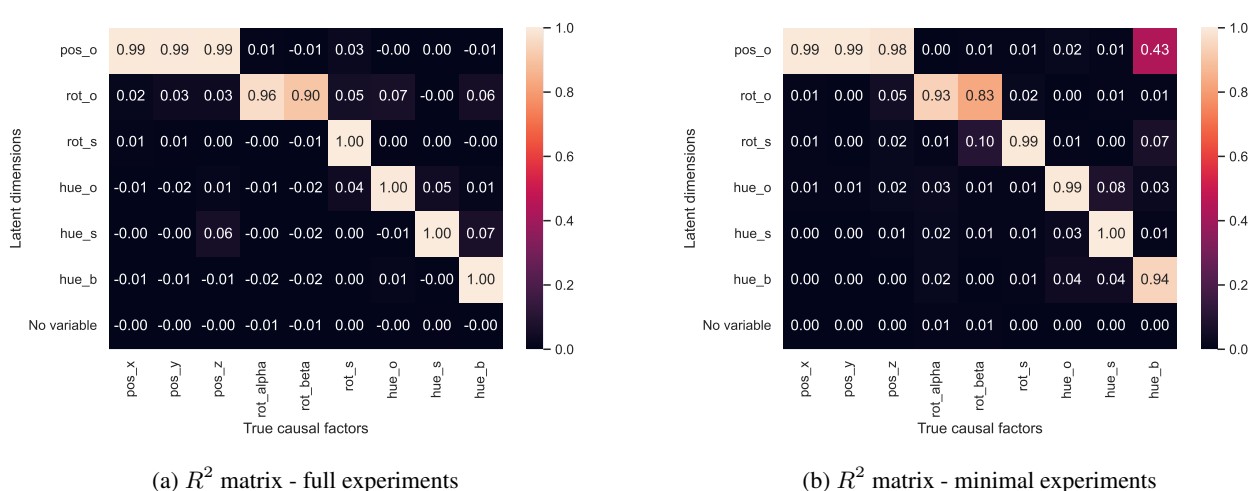

(a) $R^2$ matrix - full experiments

(b) $R^2$ matrix - minimal experiments

Figure 3: Correlation matrices of CITRIS-NF for example experiments on the Temporal Causal3DIdent dataset. The y-axis shows the sets of latent dimensions that were assigned to a certain causal factor. The set $z_{\psi_0}$ is represented by 'no variable' in the plots of iCITRIS. The x-axis shows the ground truth causal factors with all dimensions, *i.e.* pos_o represented by pos_x, pos_y, pos_z. The heatmap is the correlation matrix between those factors. **Left**: The results of CITRIS with the full set of experiments (taken from Lippe et al. (2022b)). **Right**: An example result of CITRIS on the minimal set of experiments. This experiment was conducted in the experimental setting 1 (Figure 2a).

able, there exists an experimental setting where it is intervened in only one experiment, and that there exists a different experimental setting where it is intervened in two experiments. For instance, pos_o is intervened in only $E_2$ for the second experimental setting (Figure 2b), and intervened in $E_2$ and $E_4$ for the first experimental setting (Figure 2a).

## A.2 ADDITIONAL RESULTS

We provide examples of the $R^2$ correlation matrices between learned latent variables and true causal factors in Figure 3. The left figure shows the result of CITRIS when trained on the full experiment setting, *i.e.* intervention targets are independently sampled. The figure on the right, on the other hand, shows the correlation matrix for a minimal

set of experiments, specifically the setting of Figure 2a. As mentioned in Section 4, the model is still able to disentangle the variables well, but additional correlations between variables with similar intervention patterns can occur. The correlation of 0.43 between pos_o and hue_b is one example for such, since pos_o and hue_b only differ in $E_2$ in terms of interventions.