# OpenReview forum: "Intervention Design for Causal Representation Learning"
_auai.org/UAI/2022/Workshop/CRL — CRL@UAI 2022 Poster_

### Official Review · Reviewer_tXWH · 2022-06-28

**Rating:** 6
**Confidence:** 4

**Review:**

The paper presents an upper bound on the number of experiments needed to identify
the latent causal variables from high-dimensional observations.
This work operates under the assumptions of [1], i.e., where the underlying latent variables are caused by
the variables at the previous time step and there are no instantaneous causal effects.
The work presents an upper bound for the number of experiments needed to satisfy
the assumptions that [1] requires (since the work uses the method of [1] for learning the representations).
The key contribution is to show that (\floor{log(K)} + 1) experiments are sufficient
if we had the ability to intervene on the hidden variables and do so on multiple
variables simultaneously.

I think the contribution is fairly incremental
(since the work heavily relies on previous work) which makes this a borderline paper.
But my recommendation is to accept since this work fits well with the theme of
the workshop.

A few questions:

Hard vs soft interventions:
The writing is a bit imprecise and it is not clear whether the work deals with hard or soft interventions
(or does it not matter). It would be good if the authors can spend a little more time in the introduction
or preliminaries explaining precisely what kinds of interventions they are dealing with.

Comparing with existing bound:
The authors mention that \floor(log(K)) + 1 are required in the standard causal discovery case. However, this is
only true for hard interventions. For soft interventions, just a single experiment is sufficient if multiple
simultaneous targets are allowed (see Theorem 2 in [2]). If soft-interventions are allowed, can a single intervention
be used similarly? Why or why not?
The comparison to previous work is unclear. It is not obvious what makes this setting particularly different given that even in this work,
we are intervening on the hidden variables. So it is not clear to me why additional intervention(s) are needed to begin with.

Equation (1) and the four conditions:
In Section 3.2., the work states Equation (1) and says that it can be written in the form of
four conditions. It is not clear why the four conditions are equivalent to Eq. (1).
It seems like they are sufficient, but not at all necessary. As shown in Fig. (1), as long as there is an
edge between I^{t+1}_i -> C^{t+1}_i, Eq(1) is guaranteed to be satisfied (assuming I^{t+1}_i is not a degenerate random variable).
This also relates to my previous point on using soft-interventions: can you get away with using some constant number of experiments?

I think it would be useful to spend more time talking about the four conditions since Algorithm 1 essentially
constructs a set of experiments that satisfies those 4 conditions.

"minimal causal variables":
I did not fully understand the sense in which this representation is "minimal". By minimal, do you mean
the set of causal variables that get intervened on? Is the value of K known in advance?

lower bounds:
In this work, an upper bound has been shown on the number of experiments. What about a lower bound?
Are there causal graphs where these number of experiments are necessary?

[1] Lippe, P., Magliacane, S., Löwe, S., Asano, Y. M., Cohen, T., & Gavves, E. (2022). CITRIS: Causal Identifiability from Temporal Intervened Sequences. arXiv preprint arXiv:2202.03169.

[2] Eberhardt, F., & Scheines, R. (2007). Interventions and causal inference. Philosophy of science, 74(5), 981-995.

---

### Official Review · Reviewer_36Qx · 2022-06-29
**Good workshop paper with clear result at the intersection of experiment design and CRL**

**Rating:** 8
**Confidence:** 3

**Review:**

**Summary**: The authors present the minimum number of experiments that are sufficient to identify causal variables in the framework of CITRIS, a recent causal representation learning method for data with temporal intervened sequences. The result is a bound of $\lfloor(\log(K))\rfloor+2$, where K is the number of causal variables, which is just one more than a classical bound in standard causal discovery.

**Comments:** The paper is well written and has a clear outline. There could have possibly been more space for background on the CITRIS model, as this is a clear prerequisite for this work, but it was nonetheless possible to follow the reasoning and the derivation to arrive at the main result of $\lfloor(\log(K))\rfloor+2$ experiments. However, I had also read the original work previously.

Some more explanations should be added about what kind of interventions are exactly needed (e.g. what if the intervention does not change the distribution?)

As the authors explain, this bound is similar to the one in classical causal discovery and has thus a lot of potential for extensions in the future. The result is a first of its kind at the intersection of experiment design and causal representation learning, and therefore should definitely be presented at this workshop.

---

### Meta-Review · Program_Chairs · 2022-07-06

**Recommendation:** Accept (Poster)
**Confidence:** 4

**Metareview:**

Both reviewers agree that the paper fits the scope of the workshop well and is worth presenting, so I recommend acceptance. The authors are encouraged to consider reviewer `tXWH`'s remarks about the relationship to soft interventions and lower bounds/necessary conditions.

---

### Decision · Program_Chairs · 2022-07-06

Accept (Poster)